# IMAGE-IN: Interactive web-based multidimensional 3D visualizer for multi-modal microscopy images

**Yubraj Gupta**[1,2]*, **Carlos Costa**[1], **Eduardo Pinho**[3], **Luís A. Bastião Silva**[3], **Rainer Heintzmann**[2,4]

**1** Department of Computer Engineering, University of Aveiro, Aveiro, Portugal, **2** Leibniz-Institute of Photonic Technology (Leibniz-IPHT), Jena, Germany, **3** BMD Software, Aveiro, Portugal, **4** Institute of Physical Chemistry and Abbe Center of Photonics, Friedrich-Schiller-Universität Jena, Jena, Germany

* ygupta@ua.pt

**Data Availability Statement:** All relevant data are within the paper.

**Funding:** This work was supported by the Marie Sklodowska Curie ITN-EID, Horizon 2020 project

## Abstract

Advances in microscopy hardware and storage capabilities lead to increasingly larger multi-dimensional datasets. The multiple dimensions are commonly associated with space, time, and color channels. Since "seeing is believing", it is important to have easy access to user-friendly visualization software. Here we present IMAGE-IN, an interactive web-based multi-dimensional (N-D) viewer designed specifically for confocal laser scanning microscopy (CLSM) and focused ion beam scanning electron microscopy (FIB-SEM) data, with the goal of assisting biologists in their visualization and analysis tasks and promoting digital work-flows. This new visualization platform includes intuitive multidimensional opacity fine-tuning, shading on/off, multiple blending modes for volume viewers, and the ability to handle multi-channel volumetric data in volume and surface views. The software accepts a sequence of image files or stacked 3D images as input and offers a variety of viewing options ranging from 3D volume/surface rendering to multiplanar reconstruction approaches. We evaluate the performance by comparing the loading and rendering timings of a heterogeneous data-set of multichannel CLSM and FIB-SEM images on two devices with installed graphic cards, as well as comparing rendered image quality between ClearVolume (the ImageJ open-source desktop viewer), Napari (the Python desktop viewer), Imaris (the closed-source desktop viewer), and our proposed IMAGE-IN web viewer.

## Introduction

Much of our understanding of the cellular world is based on the study of microscopy images. The samples are studied by researchers to better understand how cellular organisms operate. Historically, scientists used simple or compound microscopes to inspect and sketch the structure of cellular organisms [1]; however, with later advances in computing power and the development of modern microscopes, it became possible to automatically capture the structure of cellular organisms in a digital format and save it in two-dimensional (2D) digital data files. Several 2D visualization software programs, compatible with typical personal computers, have

IMAGE-IN (grant agreement No 861122). NO -:
The funders had no role in study design, data
collection and analysis, decision to publish, or
preparation of the manuscript.

**Competing interests:** The authors have declared
that no competing interests exist.

been developed and made publicly available to aid researchers and pathologists over the previous two decades [2].

Recent major advances in computer hardware modules (particularly in graphics processors) have increased the imaging throughput in microscopy. Automatic, fast, and reliable collection of larger datasets and hardware module-based image analysis during acquisition [3] are now standards. High-resolution digital images with many dimensions can be generated almost indefinitely by contemporary scanners [4, 5].

Deconvolution-based Widefield as well as CLSM systems [6–8] routinely generate crisp three-dimensional multichannel time-series data of fluorescence labelled targets in cells [7]. Particularly, CLSMs excel at collecting stacks of optical slices from hundreds of microns thick specimens to generate three-dimensional (3D) images, as well as (4D) time series of living cells and tissues. Here 3D/4D refers to the acquisition of digital samples of objects scattered throughout 3/4-space, i.e., in x, y, z, and t dimensions, typically but not always with isotropic spacing.

Multidimensional images can have more than three dimensions, and the order of the dimensions depends on the scanner. For example, the CLSM scanner can generate 5D images with the dimension order XYZCT (where C and T stand for channel and time). Multiple colour channels (C), particularly those acquired in different microscopy modes [9], could be of a fifth dimension. In the context of the channel, a colour is assigned to each element of an image (pixel), which is represented by a multichannel spectrum. In the case of confocal fluorescence microscopy, each spectral channel can additionally include an array of temporal components that provide a histogram of the fluorescence signal's excited state lifespan at a specific wavelength. Likewise, biologists and pathologists found that high-quality multidimensional images of specimens are very valuable for examining the intricate inner structural behavior of a subcellular organism, which was previously difficult to investigate [10]. High-quality visualization software is therefore paramount to exploring the complicated interiors of these species. To assist microbiologists in visualizing obtained high-quality multidimensional images, a number of closed [11] (e.g., Zeiss Zen, Amira [12], Arivis, Imaris, Huygens, Image-Pro, and others) and open (e.g., ImageJ [13, 14], ClearVolume [15], Vaa3D [16, 17], Blender, VTK, Drishti, Napari [18], and others) source 3D rendering applications have been developed. These visualizers not only allowed them to analyze the image in 3D but also to better understand the insight tale of each multicellular multidimensional data, which was previously impossible to get while viewing the same image in a 2D visualizer.

All of the microscopy tools mentioned above are operating system (OS) based, and the majority of them read image files into the system using the OME-Bioformats [19] library. Closed-source tools are not freely available to the microscopy community, but open-source are; however, interactive features (such as mouse movement, wand motion, keypress, and so on) are inferior to those of closed-source tools. The problem with all closed-source proprietary tools is their inability to support multiple data file formats (rather than targeted file formats) as well as the adoption of restrictive licensing agreements.

Open-source tools are better suited for the development of new analytical tools since they are free to use and do not have restrictive license requirements. Most of the closed commercial tools, such as Imaris and Arivis, do not support multiple platforms, whereas most open-source tools support all three major OS platforms (Windows, Mac, and Linux). Publicly available software could be extremely useful for the average user (even though some tools are more difficult to install on one's own PC), even if its graphical user interface (GUI) is inferior to that of commercial software.

Based on the aforementioned, we can infer that both open and closed-source OS visualizers have some limitations in their tools (feature limitations in the case of open-source tools and

platform limitations in the case of closed-source tools), even though both provide excellent visualization techniques. Therefore, we believe there is a need to develop a user-friendly visualizer for the microscope community that can bypass the high-end hardware requirements and installation procedure while giving the same degree of high-quality 3D rendering techniques and capabilities seen in OS-based visualizers.

Taking recent advancements in web browser rendering frameworks such as OpenGL [20], WebGL [21, 22], and the upcoming WebGPU [23] for displaying 2D and 3D vector graphics into account, some multidimensional viewers such as Kaibu (https://kaibu.org/) have been developed that can be used to visualize and annotate multidimensional images on the web. It's a great web tool for rendering single-colour channel images; however, this tool has limitations when visualizing multichannel images (CLSM images). It also does not support isosurface rendering.

Because of constraints in both the operating system and the web-based viewer, we were motivated to create JavaScript-based interactive 3D multidimensional visualizers for 5D microscopy datasets. As a result, in this paper, we offer IMAGE-IN [24], an interactive web-based 3D visualizer for the visualization of multidimensional images obtained from CLSM and FIB-SEM microscopes. The purpose of the creation of the IMAGE-IN 3D visualizer is to give users a fast, efficient, and flexible method to display multidimensional imaging data without worrying about hardware requirements or the installation/building process. Furthermore, the main rationale for proposing a web-based visualizer is that it allows academics to view multidimensional images from anywhere and at any time. This implies they won't have to worry about hardware or platform requirements and can run on any platform web browser with internet access.

In our IMAGE-IN [24] web viewer, we have implemented four different viewing modes: volume, surface, tri-planar, and multi-planar rendering. We assessed the proposed multidimensional visualizer's performance by loading and displaying multidimensional confocal microscope images as well as a series of focused-ion beam electron microscope images, as well as comparing rendered image quality with OS-based ClearVolume (the ImageJ open-source desktop viewer), Napari (the Python desktop viewer), and Imaris (the closed-source desktop viewer) visualizers.

## Materials and methods

Fig 1 depicts the overall proposed framework, which includes the development of a proprietary file conversion pipeline (task-1), a 3D multidimensional high-resolution image visualizer (task-2), and the creation of an automatic content discovery (task-3) APIs for the purpose of segmentation and classification of microscope data. Previously, we have already proposed a pipeline for converting proprietary high-resolution microscope image files into standard DICOM [25], and we are now focused on creating a 3D multidimensional visualizer to visualize the converted files from task-1.

### Domain goals

Based on the requirements specified by our collaborators, we designed our pipeline for the visualization of multidimensional microscope data received from CLSM and FIB-SEM scanners. We have identified three goals that will assist our collaborators in visualizing multidimensional data:

**G1: Conversion of proprietary microscope data into the DICOM standard for interoperability.** Despite major technological advances in biological microscopy, this field still faces significant challenges in data interpretation and interchange due to the large variety of digital

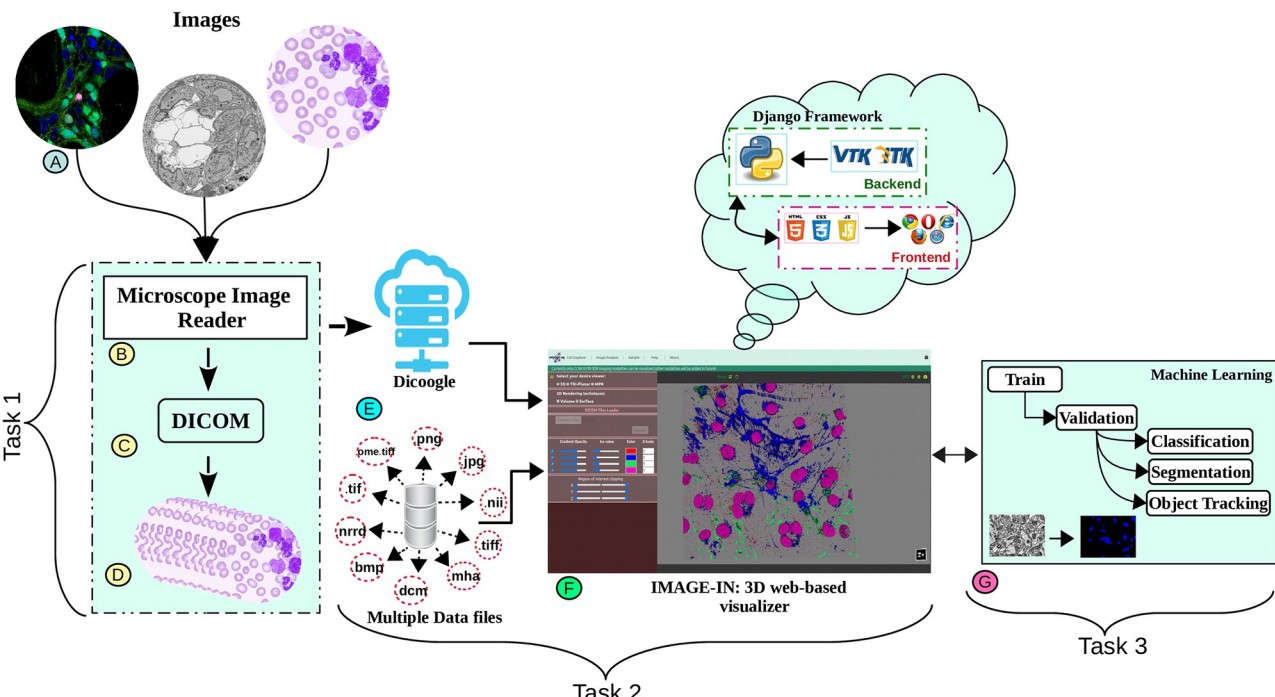

**Fig 1. The overall framework.** Task-1 includes the development of a pipeline for converting proprietary high-resolution microscope imaging modalities data into DICOM standard format. Task-2 includes the development of a 3D multidimensional high-resolution microscope image viewer. Task-3 includes the development of automatic content discovery for microscope imaging modalities, specifically CLSM and FIB-SEM images.

image formats in use. This is due to the fact that microscope images do not have globally approved file formats. As a result, a wide range of extension files are in use in the area of microscopy. Numerous open-source and corporate software tools for decoding these unique proprietary digital file formats have been developed to address the issue of multiple digital file formats. These software packages vary in terms of their intended application, usefulness, and source code accessibility. Furthermore, most offered solutions (e.g., OME-Bioformats, Open-slide, Czifile, and Tifffile libraries) only allow access to image pixel data, whereas most of the clinical data of specimens is inaccessible; additionally, as more competing supplier solutions emerge, the number of proprietary file formats grows, posing a challenge to interoperability and maintainability. As a result, there is a great demand for data standardization in the microscope field in order to improve clinical integration and support computational development streams.

To solve this issue, previously we have proposed a conversion pipeline (task-1) [25] that takes several distinct proprietary file formats as input and converts them into standard DICOM files. We have tested the performance of the proposed method by passing several distinct proprietary datasets belonging to Zeiss, Hamamatsu, Olympus, Nikon, Leica, and other manufacturers. Furthermore, to determine whether the generated DICOM files were readable or not, we read each DICOM file in a Python environment using the Pydicom library, and we passed each generated DICOM file to the open-source PACS archive Dicoogle [26] to be opened by the PACScenter viewer, the results of which can be found in our previous article [25].

**G2: Visualization of multidimensional microscope imaging data.** In the context of visualizing multidimensional 3D images, most prior built 2D visualizers use algorithms such as cross-sectional (tri-planar: XY, YZ, XZ views) and multi-planar reconstruction (MPR) to

display a 3D stack of images. Those methods slash the volumetric data at an arbitrary angle and show the 2D slices cut on the cross-section plane. They are still extremely popular among biologists to visualize 3D data due to their simplicity and efficient computing power input [28], being used in scientific visualization packages such as ImageJ (Fiji) [13, 14], Vaa3D [16, 17], Imaris, and others. In actuality, the tri-planar visualizer only displays the 3D data's outer surface in three cross-sectional planes and cannot depict the underlying structure of the sub-cellular organism and also viewing a two spatial dimension image is not difficult because it can be viewed on any computer using an appropriate imaging program, but viewing a three spatial dimension image requires the use of special software. Therefore, to fully comprehend the inner structure of the intricate subcellular organism, a biologist requires a multidimensional 3D viewer. It provides both quantitative and qualitative information about an object or object system using digital images collected through various modalities [29]. Fig 2 shows the multidimensional multichannel CLSM microscope image. The 3D viewer, along with interactive models that allow scientists to move and rotate objects, will make it much easier for biologists to reveal new insights into cellular organisms.

**G3: 3D visualization techniques.** Previously, multidimensional images were commonly viewed in a cross-sectional viewer as a collection of individual slice planes; however, with recent advancements in visual graphics hardware, they are now frequently integrated using different projection methods (such as ray casting or voxel projection) to generate a single

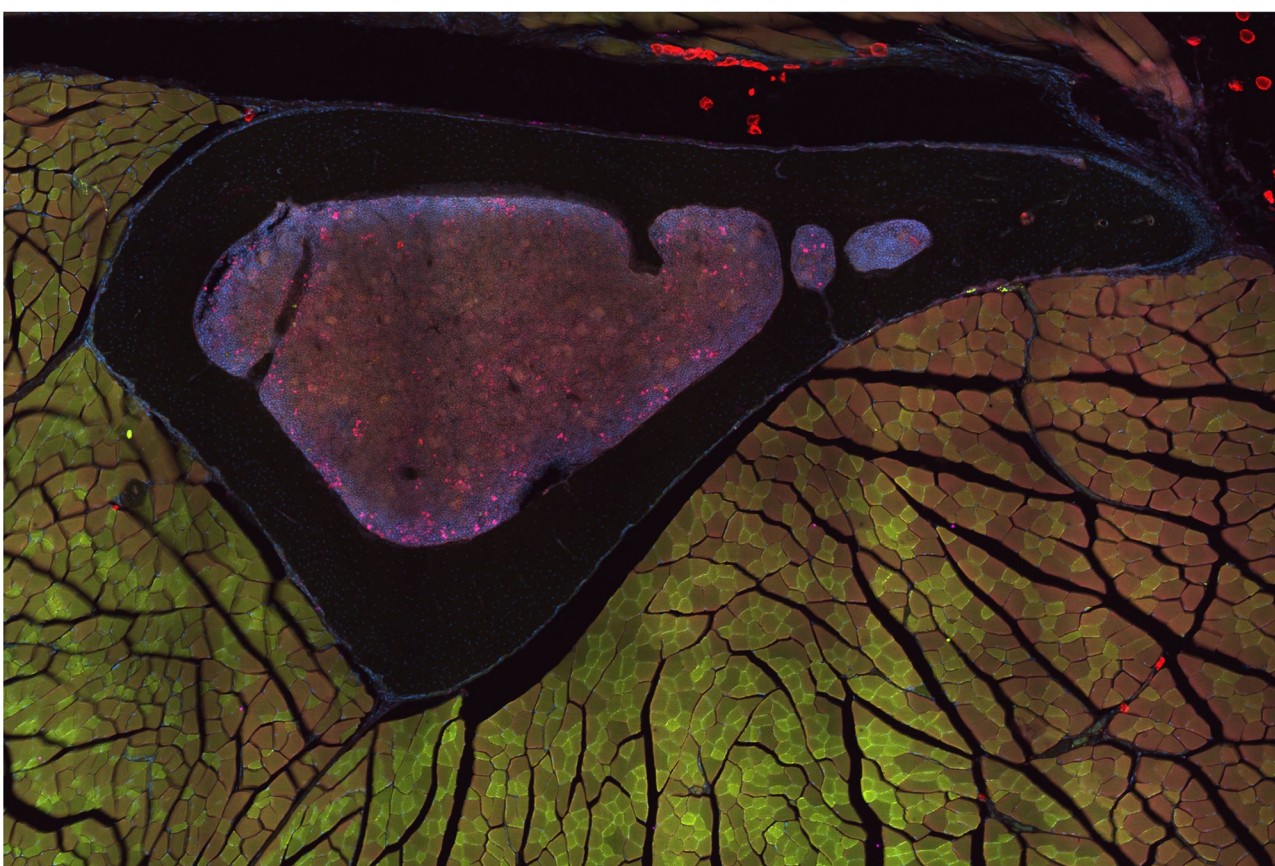

**Fig 2. The four-channel tibia bone [BL6-SEC5] of mice [27] (Perilipin 1 (visualized in red), Tyrosine Hydroxylase (visualized in green), DAPI (visualized in blue), calcitonin gene-related peptite (visualized in pink)).**

display object (as shown in Fig 1(F)) onto the 2D screen at interactive speed, allowing the user to examine the data from any chosen angle and zoom factor. The rendering speed is achieved by using powerful hardware graphics processing engines and thorough software optimization. These projection techniques can be further combined with other computer graphics techniques such as basic geometric models, reflection, and shading to generate a realistic 3D image from the microscope image.

(a). **Surface rendering**

Surface rendering [30, 31] methods can be used when just the outer contour of a 3D object has to be properly depicted. Isosurface extraction ("Marching Cubes") [32, 33] is a popular method for to visualizing 3D image data on a 3D surface. It works by extracting a 2D surface mesh from a 3D volume. Volumetric image data are made up of stacked 2D images and can be viewed as a three-dimensional array matrix of image data points. A volume element (voxel) is the smallest piece of a 3D dataset of a 3D image, similar to a pixel in a 2D image. The 3D surface is extracted by marching across the entire volume using a threshold-based technique for each voxel cube of the lattice. Each linked corner in each voxel cube is searched for a threshold-crossover point using a linear interpolation algorithm. The isosurface within each voxel cube is formed by connecting these points along each of the edges, and the procedure is repeated throughout the volume. When dealing with samples that correspond to numerous surfaces inside the same voxel cube, special processing is necessary. Surfaces are often represented as triangles. This approach has the benefit of producing a highly comprehensive surface representation for the objects of interest, which is especially useful when the objects' structure is clearly distinguished by their signal-to-noise distribution in the data. In surface rendering, object of interest are rendered dynamically straight from the scene, therefore, rendering speed is critical in this method. Because this algorithm is computationally expensive for the graphics hardware, the generation of the new 3D surface is delayed each time a new threshold (or isosurface) value is set. This approach generates a significant number of triangles in a typical collection of volume image data, generally in the tens of thousands. As a result, displaying them all can be an intensive graphic task. Instead of threshold intervals, any automated, rigid, boundary, or region-based approach can be employed, and the pace of the segmentation and rendering process must be adequate to make this form of viewing viable.

(b). **Volume rendering**

Volume rendering [22, 34] methods can be used when the interior of a 3D object needs to be displayed. The main limitation of surface extraction methods is that they represent a fixed iso value in the data, which makes them selective or restricted. As a result, this approach is computationally expensive for the hardware to generate a new surface when a new iso value is set. Volumetric visualization approaches transcend these limits and may aid in rendering as much information as possible from a given 3D volume in a much faster way.

At the moment, there are several volume rendering methods available, but maximal (or minimal) intensity projection (maxMIP or minMIP) and alpha-value blended views are the two most popular types of methods for displaying 3D data in volumetric visualization [35]. In the MIP approach, the maximum pixel intensities of the volume data are projected on the viewing plane along each ray of the projection. This approach is extremely beneficial for displaying high-intensity patterns of volumetric data. The MIP volume rendering method does not provide shading information of an object, therefore image depth and occlusion are not evident.

To circumvent this constraint, a blending procedure similar to that employed for anti-aliasing during scene composition or scan conversion is being used [36]. The blending procedure is equivalent to taking the weighted average score mean of the data point and the background value. The weighting factor (or opacity value) might be given a physical meaning by the data point's "opacity" from which the data point's transparency can be derived. The basic idea behind the blending volume rendering procedure is to give each voxel in the scene an opacity range from 0% to 100% which enhances or suppresses their influence in the rendered image. The opacity level is decided by the objectivity value at the pixel and how strongly this specific grade of objectivity is to be depicted in the rendering. The purpose is to figure out how much light gets to each voxel in the viewing plane. The opacity of the pixel determines the quantity of light transmitted. Light emission is proportional to objectness and opacity: the stronger the objectness, the larger the emission. The transfer function, which represents a lookup table, can be used to automatically control the transparency or opacity of the rendered image. An intriguing element of this method is that modern graphics cards available today offer per-pixel blending functions, which are also employed for spatial anti-aliasing effects and scene composition.

## Visualization toolkit (VTK) and insight toolkit (ITK)

VTK and ITK are two of the most well-known open-source bioimaging libraries for developing scientific visualization applications and applying image analysis/processing functions to input data [21]. VTK and ITK are both structured as a collection of sources (data processing units), filters that accept input data, mappers, and finally filters that offer processed output data. Readers and writers are both common instances of sources and mappers. Readers feed data into the pipeline (usually from a file), while writers emit data. Because of the versatility and adaptability of these two class libraries, users can combine numerous filters into processing frameworks to address unanticipated processing requirements. ITK and VTK were originally written in C++ and were eventually wrapped in various languages, most notably Python, Java, and Tcl/Tk. In addition, since 2017, both of these tools have been available in JavaScript for web scientific visualization and web image processing. Both programs assist in the management of datasets that are too large to store in the main computer memory by dividing incoming data into small tiles that are then processed one at a time.

VTK offers a broad range of scientific rendering techniques for 2D and 3D geometric models, such as vector, scalar, texture, tensor, and volumetric approaches, as well as sophisticated modelling methods like implicit modelling, mesh smoothing, polygon reduction, contouring, and so on. ITK is a companion package that focuses on data analysis/processing rather than rendering; application developers usually employ ITK and VTK together to produce robust applications for the user. ITK is one of the most comprehensive image-analysis method packages, notably for image registration, segmentation, stitching, and extraction of a feature. ITK, like VTK, supports n-dimensional image analysis. ITK also supports a wide range of image file formats, including BigTIFF, JPEG 2000, HDF5, and others.

Recently, VTK developers have proposed a VTK.js [21, 31] package, which is a re-implementation of the popular VTK/C++ visualization toolkit into the JavaScript (ES6) language to render objects into web browsers. VTK.js makes use of OpenGL and WebGL technologies to render geometric and volumetric objects onto the web using local computer graphics hardware.

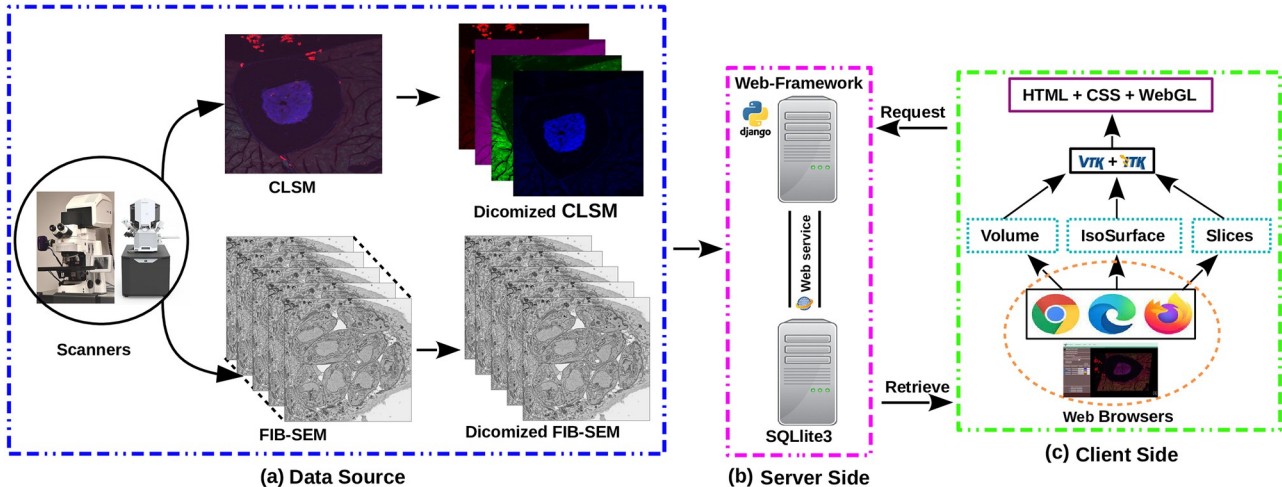

**Fig 3. Framework for an interactive 3D viewer.** (a) Acquisition of multidimensional images and conversion to the DICOM standard; (b) Development of a web server using the Django web-framework; and (c) Implementation of visualization techniques in the client-side framework.

## Overview of the proposed 3D multidimensional visualizer architecture

Fig 3 depicts the conceptual design of the 3D visualization framework. The proposed 3D multidimensional visualization framework, like other web application packages, is a server-client-based system structured in a three-tier design: a data source layer (input), a server-side logical layer (backend), and a client-side application layer (frontend), the functions of which are detailed below.

(a). **Data source layer**:

The imaging data (here CLSM and FIB-SEM) displayed in this 3D multidimensional visualization framework can be acquired from various confocal and focused-ion beam scanning electron microscopes. Our IMAGE-IN web-viewer can currently render input images belonging to.dcm, .tif, .tiff, .nrrd, .mha, .nii, .ome.tif, .png, .bmp, .jpg, and other formats; however, due to the unavailability of a generic import tool in JavaScript, the IMAGE-IN web-viewer is still unable to load proprietary files such as.czi, .lif, .nd2.

(b). **Server-side application layer**:

The integrated server-side component consists of a web server and a database server. The web server is a middleware layer that hosts all specified built web services that make use of system functions. These backend web apps, developed in the event-driven Django web framework (version 3.2.8) environment (is a high-level Python web application framework that supports the rapid creation of clean APIs with a pragmatic architecture), interface with the DBMS for data retrieval, parsing, accessing, and publishing. On the web server side, a Gunicorn (WSGI) HTTP server was configured to handle HTTP requests and responses, whereas on the database server side, an SQLite database was used. SQLite's databases are considered to be an ideal choice for developing read-only apps, such as ours, where the backend is solely utilized to read input data.

(c). **Client-side application layer**:

This is a comprehensive web interface that combines a graphical user interface (GUI) and the primary 3D rendering canvas, allowing user interaction for accessing data, interactive multidimensional 3D visualization, and analysis. The frontend interfaces were

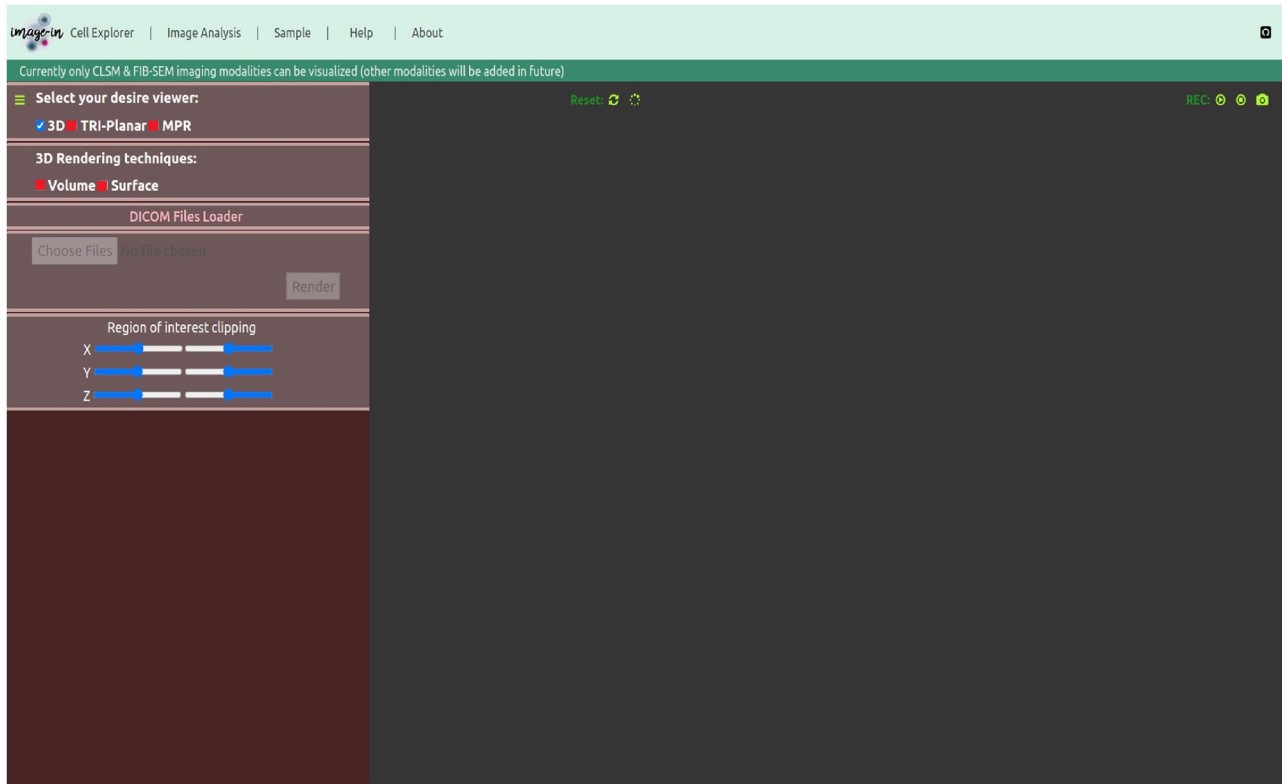

**Fig 4. Graphical user interface (GUI) of the IMAGE-IN multidimensional 3D visualizer for visualizing CLSM and FIB-SEM data.**

created in a Pycharm programming environment utilizing HTML5, JavaScript, VTK.js (version 19.2.12), ITK.js (version 14.0.1), jQuery, and cascading style sheet (CSS) libraries.

Our proposed interactive multidimensional 3D visualizer supports four visualization modes, as shown by the red blur box in Fig 4. The three-dimensional option includes two 3D viewers: volume and surface rendering, as well as two more visualization modes: triplanar and the MPR viewer. The WebGL-based visualization toolkit (VTK.js) package was used to render a series or 3D stack of multi-format input images into the HTML5 canvas.

**Implementation.** Our multidimensional web visualizer is built utilizing packages such as the Django web framework (for server development) [37], ITK.js (for reading multiple input files) [38], and VTK.js (for visualization) [21, 31]. The main source code of our visualizer (to run it locally) has been made public and can be found at the following GitHub link [24]. Furthermore, users can test the IMAGIN-IN visualizer by visiting the following URL: https://yuvi. pythonanywhere.com/, hosted on the domain pythonanywhere.com. The documentation for using the IMAGE-IN visualizer can be found at GitHub link [39], demo videos [40] and the sample can be downloaded from GitHub link [41]. Please keep in mind that the IMAGE-IN web-viewer presently renders input files in the following formats:.dcm, .tif, .tiff, .nrrd, .mha, . nii, .ome.tif, .png, .bmp, .jpg, and many others.

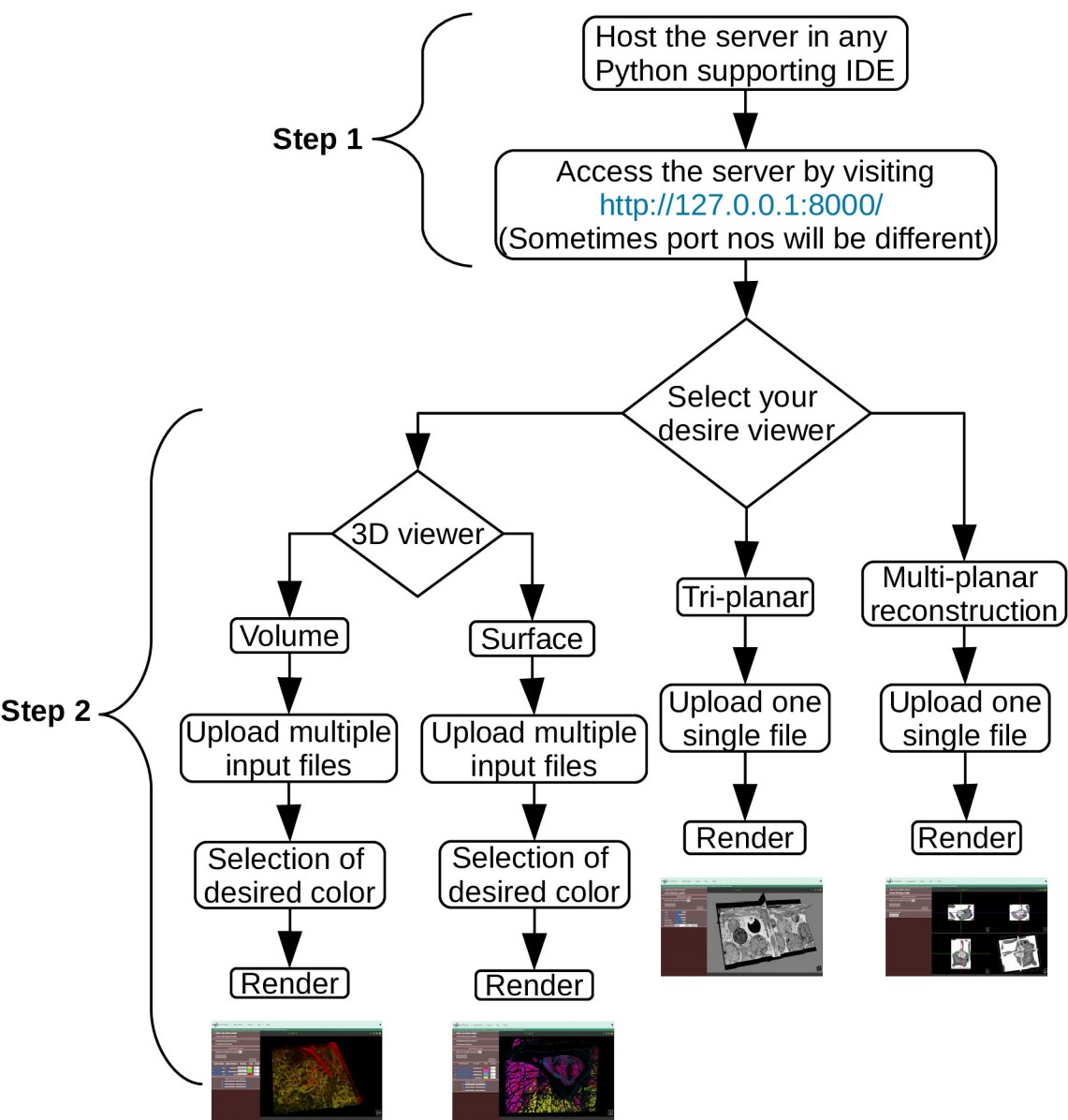

**Fig 5. Flowchart depiciting the inner working mechanism of our viewer.** Step 1 describes how to host the server locally, and Step 2 explains how the viewer works, from describing the desired viewer to clicking the render button.

**Working mechanism.** The working mechanism of our proposed viewer is divided into two main steps, as shown in the graphical flowchart in Fig 5.

In the first step, users have to run the server locally on their laptop/PC device in any integrated development environment (IDE) software that supports the Python programming language, this is because we have used the Python-based Django web framework for the building the web server. Therefore, the user should first host the server locally, and access the viewer by visiting http://127.0.0.1:8000/ in the preferred web browser.

After successfully running the server locally and accessing the above-mentioned link in any web browser, the user will see the visualization window depicted in Fig 4, where the previously described viewing modes (volume-/surface-rendering, tri-planar, and MPR) are now available.

Each viewing mode has its own advantages and disadvantages. These four viewing modes were created using VTK.js, which, as mentioned above, uses the OpenGL/WebGL graphics library for rendering 2D and 3D visual elements within any compatible web browser without the need for running plugins and focuses on geometry (isosurface) and volume rendering using vtkPolyData and vtkImageData classes from the native VTK environment.

The volume rendering mode uses the vtkVolume, vtkVolumeMapper, vtkPlanes, and vtkVolumeProperty classes to render VTK objects in 3D. VTK.js accepts input in the form of VTK objects and currently, it does not support or read.dcm, .tiff, .tif, .nii, or any other graphical or raster pixel image files other than.vti and.vtp (which are VTK legacy file formats); thus, we use the ITK.js library to read raster pixel images that can be in a series of images or 3D stack multidimensional image files and then internally transform those input files into VTK objects before passing them to the volume rendering pipeline for 3D volume rendering. Furthermore, we have implemented three blending methods in the volume rendering mode: composite, maximum, and minimum intensity projection, to assist users in gaining detailed insight into the visualized data. Likewise, the vtkColorTransferFunction class is utilized to set the specific color map to render the specific channel of an image volume. Moreover, we have also integrated an opacity tuning for each 3D rendering, ranging from 0% to 100%. This functionality helps enhance or suppress the light on the rendered image. Shading features are also implemented into the volume rendering viewer. It aids in the improvement of 3D model illustrations by adjusting the amount of darkness on an object. If shading is disabled, the mapper for the volume will not execute shading computations. If shading is enabled, the mapper may execute shading computations; however, in some situations, shading does not occur (for instance, in the case of maximum or minimum intensity projection), and so shading will not be done even when this flag is enabled.

Similarly, the surface/isosurface rendering application uses the vtkImageMarchingCubes, vtkActor, vtkPlanes, and vtkMapper classes to transform VTK object files into surface objects that can be rendered. Here, the vtkImageMarchingCubes filter is used to produce isosurfaces. It works by creating cuboid-shaped cells (voxels) using the image pixel values at the cube's of all eight corners. This cube "marches" over the whole volumetric dataset, subdividing space into a succession of cubes. At each phase, it evaluates if the cube's vertex is inside or outside the isosurface. The isosurface intersects edges that are next to one "inside" and one "outside" categorized vertex and can construct a triangular patch whose corners are obtained via linear interpolation along those cube edges. Later, by joining the patches from each step of the cube, we get an approximate isosurface represented by a triangular mesh. In our experiment, we utilized the vtkImageMarchingCubes filter function from the VTK.js package to generate a mesh. Contour values must be provided to the vtkImageMarchingCubes filter in order for it to produce isosurfaces; hence, in our instance, we gave a min/max scalar range of an entire image as well as the number of contours to generate a sequence of evenly spaced contour values. Furthermore, we have included opacity and iso value tuning features for each 3D rendered image to assist researchers with their data visualization requirements.

Likewise, the tri-planar and MPR rendering APIs use vtkImageSlice, vtkImageReslice, vtkImageMapper, vtkActor, vtkMapper, vtkOutlineFilter, vtkVolume, vtkVolumeMapper, and some other classes to transform VTK object files into a two-dimensional slice plane viewer and a three orthogonal planes' viewer (orthoviewer) through the dataset parallel to the three Cartesian planes.

In the second step, the user must choose (by ticking) which viewing mode(s) to use to visualize their image data. One tab will open for each selected viewing mode, allowing the user to control the distinctive features each viewing mode supports. After the selection of a preferred viewer, now the user can begin to load multiple input files (e.g., color channels) into the

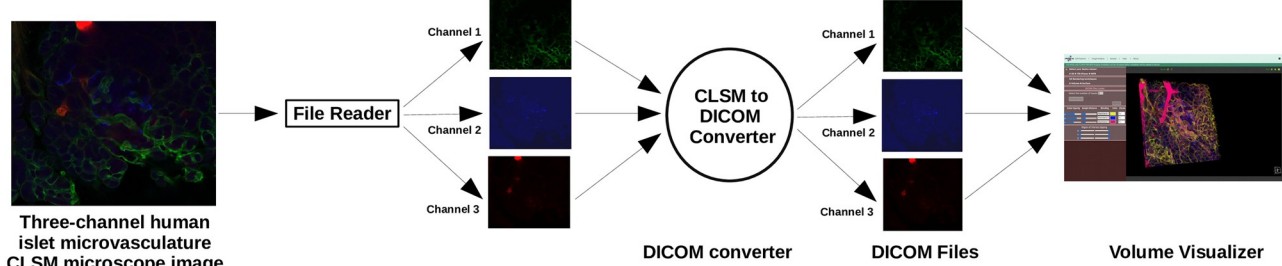

**Fig 6. Pipeline showing image conversion to volume rendering.** The OME-Bioformat library was used as a file reader to read microscope imaging raw files, and the Pydicom Python library was used to convert them into DICOM standard, which was then displayed in volume view using our proposed IMAGE-IN [24] viewer.

system. Since FIB-SEM data is typically only a single-color slice, loading FIB-SEM samples is straightforward. However, for stacked images, such as confocal datasets, each slice should be loaded sequentially.

For example, if the obtained CLSM image is a composite of three channels as shown in Fig 7(a), the user must first split each channel separately and convert each separated image file into any of the above-mentioned file formats Fig 1(E) (if it is not already in the required file formats) so that in the final stage, we will have three files/folders, each containing a sequence of slices or a stack of image files, but all of the same color Fig 6.

After successfully splitting and converting the three-channel composite CLSM image files into the appropriate formats, the user can now load them sequentially into the systems. Following each load, it is important to choose a new colour map if the user wants to display the subsequent channel in a different colour. However, it is also possible to modify it after the image has been rendered.

After finishing the loads and selecting/changing the colour maps (based on the number of inputs), the render button can be clicked. After some seconds, the canvas will display a rendered 3D volume image with three distinct colours as selected by the user, as exemplified in

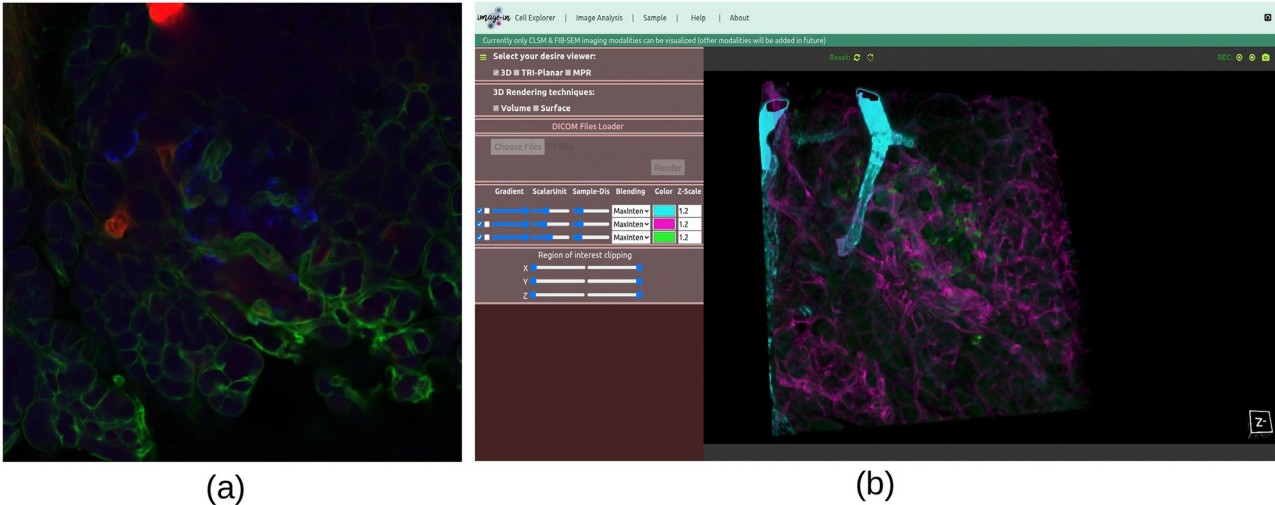

(a) (b)

**Fig 7.** (a) Example of a three-channel human islet microvasculature [42] CLSM dataset volume rendered in (b) Each channel is rendered with a different color: Channel-1 is cyan, Channel-2 is magenta, and Channel-3 is green.

Fig 7(b), where we have rendered a three-channel human islet microvasculature [42] CLSM dataset using three distinct colours, channel-1 with cyan, channel-2 with magenta, and channel-3 with green. Using the same approach, the user can visualize their image data with any other viewer.

## Result and discussion

In this paper, we present IMAGE-IN [24] an interactive web browser-based multidimensional 3D visualizer optimized for two microscope imaging modalities: namely, CLSM and FIB-SEM. The desire to develop a web-based 3D visualizer is to provide researchers with two benefits: the first is the ability to use it in any web browser without installing any third-party software or plugins into their system, which can be time-consuming to install because each package has different requirements and installation processes, and not all OS-based visualization packages work on every OS (for example, Imaris does not support Linux systems), and the second is to provide smoothness, simplicity, and efficiency while visualizing multidimensional microscope imaging modalities in the web browser.

In this section, we will present the results of evaluating the performance and reliability of each proposed 3D visualization mode by loading and visualizing multichannel CLSM and FIB-SEM microscope datasets of various input sizes (ranging from Mb to Gb in size).

### Data description and preparation for visualization

In this case study, we demonstrated the proposed visualization framework employing several CLSM and FIB-SEM microscope imaging modalities, as shown in Table 1. These raw [(.czi) human islet microvasculature [42], (.nd2) adipose tissue [43], (.tif) tibia bone tissue [27], (.lif) mouse proximal colon [44], (.lsm) submandibular ganglion [45], and (.lsm) mouse proximal colon [46]] CLSM, [(.tif) tuwongella immobilis [47], (.tif) plasmodium falciparum [48], and (.tif) parasitophorous vacuole [49]] FIB-SEM sample image files were obtained from public bio-image data sources. To assess the efficacy of the proposed visualization framework, we downloaded two-, three-, and four-channel CLSM images, each with a different image size, as well as FIB-SEM datasets of various image sizes.

The acquired raw CLSM specimens were in a composite form, which means that their channels were piled on top of each other, so each channel had a certain number of slices that were attached to each other (or were a z-stack) as shown in Fig 2, which is a four-channel Tibia bone [BL6-SEC5] tissue image obtained by a Nikon spinning disk confocal microscope and has a.tif extension file.

**Table 1. Selected CLSM and FIB-SEM microscope images.**

| Samples | Modality | No. of Channels | No. of Slices | Total Image | Image Size |
|---|---|---|---|---|---|
| Human islet Microvasculature (1) [42] | CLSM | 3 | 73 | 219 | 459.6 Mb |
| Tibia bone tissue [BL6-SEC5] (2) [27] | CLSM | 4 | 21 | 84 | 3.28 Gb |
| Adipose tissue (3) [43] | CLSM | 2 | 68 | 136 | 2.0 Gb |
| Tibia bone tissue [C3H-SEC9] (4) [27] | CLSM | 4 | 21 | 84 | 2.22 Gb |
| Tuwongella immobilis (5) [47] | FIB-SEM | . | . | 464 | 7.2 Gb |
| Plasmodium falciparum (6) [48] | FIB-SEM | . | . | 427 | 3.9 Gb |
| Parasitophorous vacuole (7) [49] | FIB-SEM | . | . | 360 | 942.7 Mb |
| Mouse proximal colon (8) [44] | CLSM | 3 | 45 | 135 | 147.7 Mb |
| Submandibular ganglion (9) [45] | CLSM | 2 | 62 | 124 | 68.2 Mb |
| Mouse proximal colon (10) [46] | CLSM | 2 | 142 | 284 | 297.8 Mb |

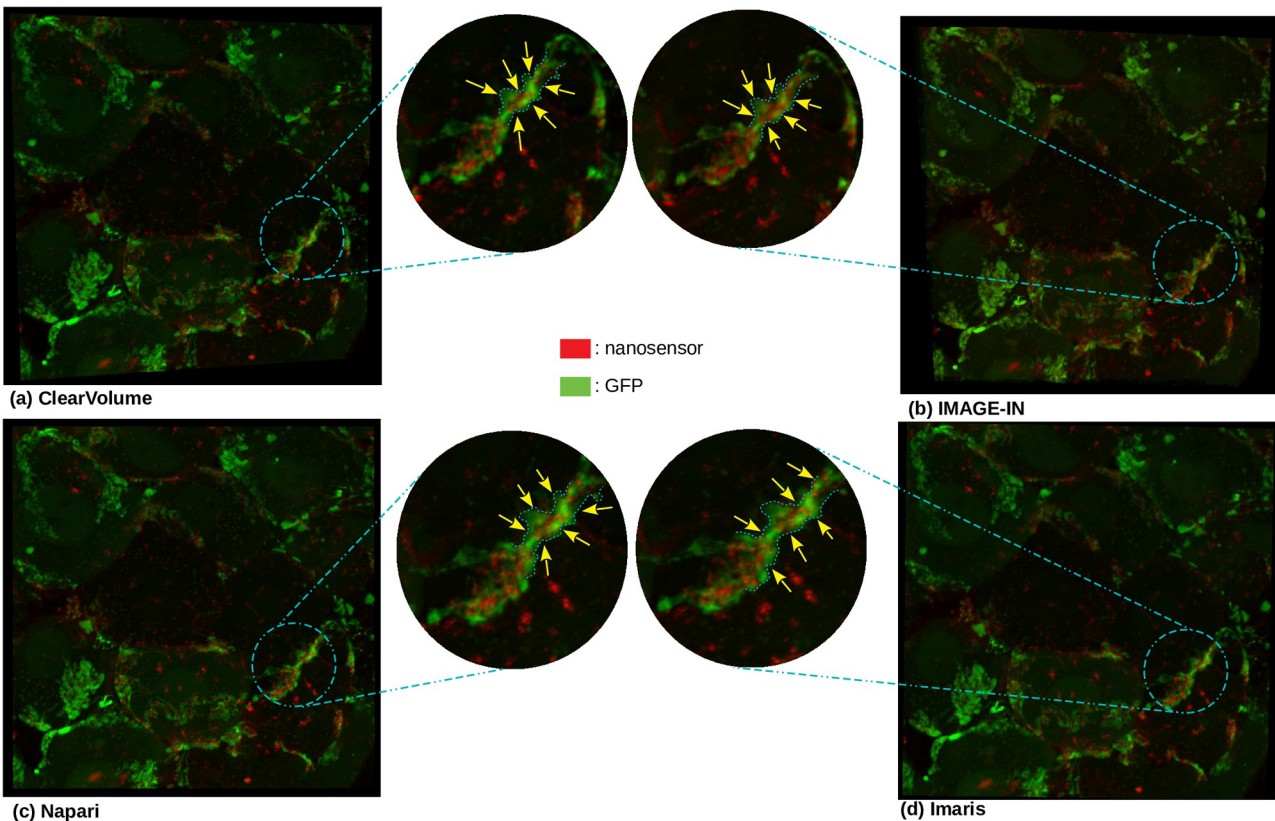

**Fig 8.** The two-channel submandibular ganglion [45] CLSM dataset is rendered in 3D volume on the following viewers: (a) Clearvolume (the ImageJ open-source desktop viewer), (b) IMAGE-IN (the web visualizer), (c) Napari (the Python desktop visualizer), and (d) Imaris (the closed-source desktop viewer). The yellow arrow on each rendered image indicates the variation in region across four viewers. The image-in viewer reveals more nanosensor (represented by red color) on this specific spot when compared to the other three viewers. This comparison is made in the early phases.

Because this is a composite image, we split each channel and afterwards convert each of them into any of the above-mentioned file formats for visualization. As with the FIB-SEM image, all samples were already in.tif format and were single-channel colour images; thus, we did not need to convert them because IMAGE-IN can read single-channel colour.tif extension files.

## Comparison between IMAGE-IN, ClearVolume, Napari, and Imaris viewers

In this section, we compared the results acquired from IMAGE-IN to the publicly available ClearVolume (a renderer running inside ImageJ) and Napari (a renderer running inside Python) viewers, as well as with the commercial Imaris (a closed-source desktop viewer) viewer. Figs 8 and 9 show the comparative results of each visualizer when two-channel submandibular ganglion [45] and three-channel mouse proximal colon [44] CLSM microscope images were passed through them. Aside from modifying mouse manipulators such as zoom and panning, these image screenshots were captured without adjusting any parameters. IMAGE-IN, ClearVolume, and Imaris viewers provide functionality such as region of interest clipping, voxel dimension scaling, an opacity slider, a color picker, and more. However, the native Napari Python visualizer does not provide region of interest clipping or voxel dimension

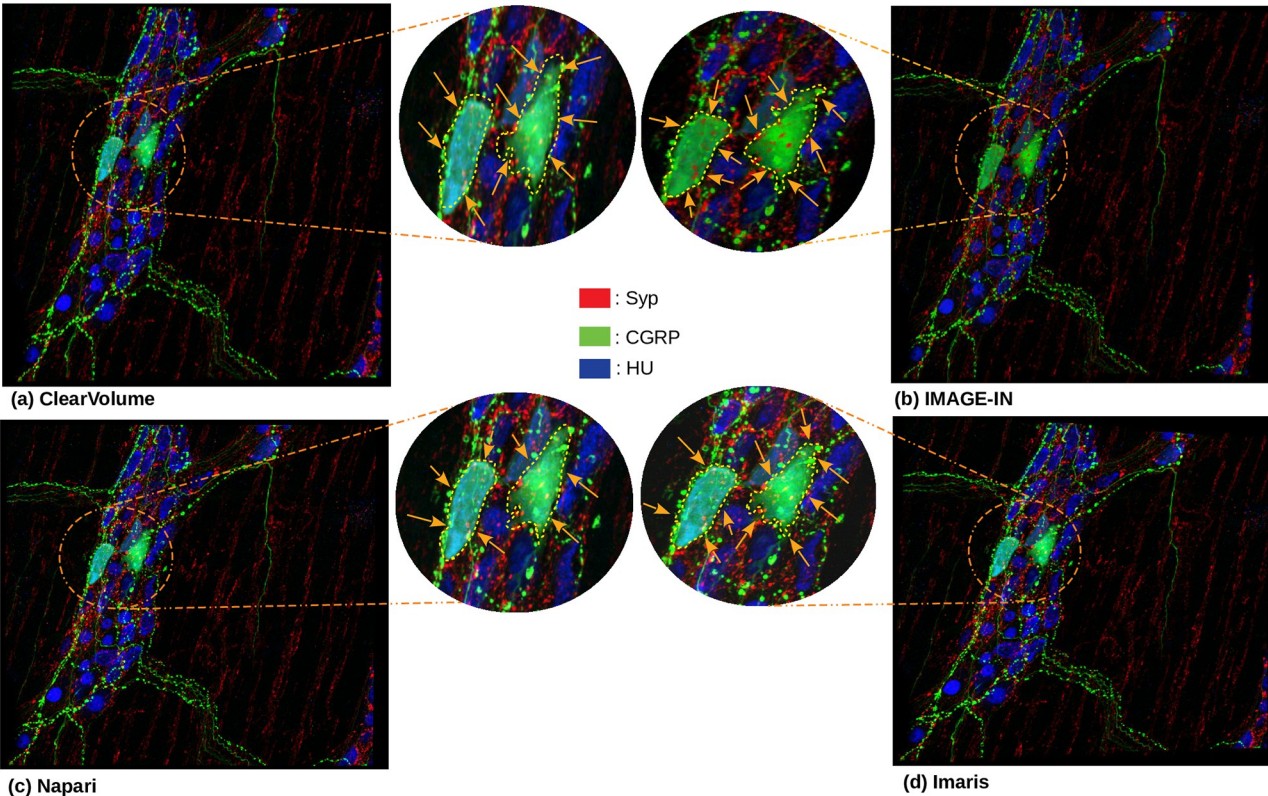

**Fig 9.** The three-channel mouse proximal colon [44] CLSM dataset is rendered in 3D volume on the following viewers: (a) Clearvolume (the ImageJ open-source desktop viewer), (b) IMAGE-IN (the web visualizer), (c) Napari (the Python desktop visualizer), and (d) Imaris (the closed-source desktop viewer). The orange arrow on each rendered image indicates the variation in region across four viewers. When compared to the other three viewers, the IMAGE-IN viewer clearly reveals more Syp (represented by the red color) and HU (represented by the blue color) on this specific spot. This comparison is made in the early phases.

scalings. ClearVolume only supports maximum intensity projection to render volumetric data; however, our visualizer offers three different blending modes to render volumetric data: composite (default), maximum, and minimum intensity projection. Napari offers five blending modes: additive, average, maximum (selected by default), and minimum intensity projection. Imaris, for example, offers four blending modes: MIP (maximum intensity projection), normal shading, blend, and shadow projection.

To render the provided input CLSM data, we use the maximum intensity projection mode in all three visualizers (Figs 8 and 9). Our visualizer provides surface (3D isosurface), tri-planar (cross-sections), and MPR (orthogonal sections) viewers. The rendered image quality is nearly identical in all viewers. The user interface (UI) in IMAGE-IN is substantially quicker than in Napari and ClearVolume viewer. However, the commercial (closed source) Imaris viewer is faster than all three viewers.

## Performance and rendering quality

Following the successful conversion of several raw CLSM and unrecognized FIB-SEM images into aforementioned file format, we ran each input file through our proposed interactive multidimensional 3D web-viewer for the visualization. Later, we evaluated the performance of the IMAGE-IN web viewer on the frontend rather than the backend because the backend is only

**Table 2. 3D viewer performance measurement on laptop and desktop devices.**

| Samples | Laptop device | | | | Desktop device | | | |
|---|---|---|---|---|---|---|---|---|
| | Loading time | | Render time | | Loading time | | Render time | |
| | VOL (ms) | SUR (ms) | VOL (ms) | SUR (ms) | VOL (ms) | SUR (ms) | VOL (ms) | SUR (ms) |
| Human-(1) [42] | 46.20 | 48.47 | 4508 | 20435.81 | 66.20 | 86.29 | 6995.89 | 17328.52 |
| Tibia [BL6-SEC5]-(2) [27] | 32.20 | 34.83 | 2471.81 | 72015.60 | 32 | 47.11 | 3455.79 | 54973.20 |
| Adipose-(3) [43] | 27.50 | 30.35 | 27939.09 | 35659 | 29.21 | 33.74 | 31247.11 | 28865.42 |
| Tibia [C3H-SEC9]-(4) [27] | 34.05 | 43.51 | 1728.01 | 55457.88 | 36.75 | 52.43 | 2547.80 | 40201.08 |
| Tuwongella-(5) [47] | 87.30 | . | 14565.78 | . | 107.37 | . | 21666.53 | . |
| Plasmodium-(6) [48] | 58.42 | . | 3321.19 | . | 66.65 | . | 5894.14 | . |
| Parasitophorous-(7) [49] | 65.36 | . | 7036.80 | . | 85.40 | . | 9948.09 | . |
| Mouse-(8) [44] | 23.10 | 24.37 | 1085 | 4176 | 26.07 | 28.41 | 1231.83 | 3548.54 |
| Submandibular-(9) [45] | 20.27 | 23.08 | 718.19 | 1114.13 | 22.53 | 23.50 | 915 | 881.33 |
| Mouse-(10) [46] | 26.81 | 28.31 | 2086.6 | 10648.66 | 27.16 | 29.46 | 2275.75 | 7846.19 |

VOL: Volume rendering, SUR: Surface rendering, ms: Millisecond.

used to host the Django web server and all other libraries (such as ITK.js and VTK.js) are loaded on the frontend when the user loads the webpage. As a result, we assess the rendering performance of the IMAGE-IN web viewer on the client side to determine the impact of the client's computer graphical power on performance. Because of this, we chose three different devices, the first of which is a laptop with a system specification (Memory: 16 Gb, processor: Intel Core i7-10750H CPU 2.60Hz, Graphics: NVIDIA GeForce RTX 2060, OS type: 64-bit, max 3D Texture size: 16384), the second of which is a desktop with a system specification (Memory: 64 Gb, processor: Intel Core i7-6700K CPU 4Hz, Graphics: NVIDIA GeForce GTX TITAN, OS type: 64-bit, max 3D Texture size: 2048), both of which were running the Ubuntu 20.04.4 LTS (Focal Fossa) operating system and both of which had a Google Chrome and Firefox browser pre-installed, and the third device is an iPad Air (3rd generation, 64 Gb), to gain a better sense of how graphical computational power will affect the end-user experience.

Table 2 compares the client-side operation times of two systems by evaluating loading and rendering times while rendering the same input sample into volume and surface 3D viewers. Please bear in mind that each duration in this section only indicates the quickest operations that can be performed while loading and displaying the provided input sample. Our viewer offers extra parameters, such as an opacity slider, shading on/off, a color picker, an iso contour value slider, region of interest clipping, z-voxel dimension scalling, and so on, which the microbiologist community may find useful and simple to use. The data loading time indicates how long it took to load the input data into the system at the start, as well as the conversion of input data from ITK to VTKObject (this was measured after the user accessed the webpage and ITK and VTK.js were already loaded into the client browser), whereas the rendering time indicates how long it took to render the image inside the viewport once the input image was loaded into the system (this was measured after the user pressed the render button, and its performance fully depends on client computational power). All timings were determined in milliseconds, and the Google Chrome web browser was used to both visualize the rendered image and well as measured the rendered times.

The rendering time is determined by numerous factors (like image resolution, number of slices, image data-types, and device processing power), but mainly if we have powerful computing hardware, then it is quite quick; otherwise, it takes a few seconds more to render.

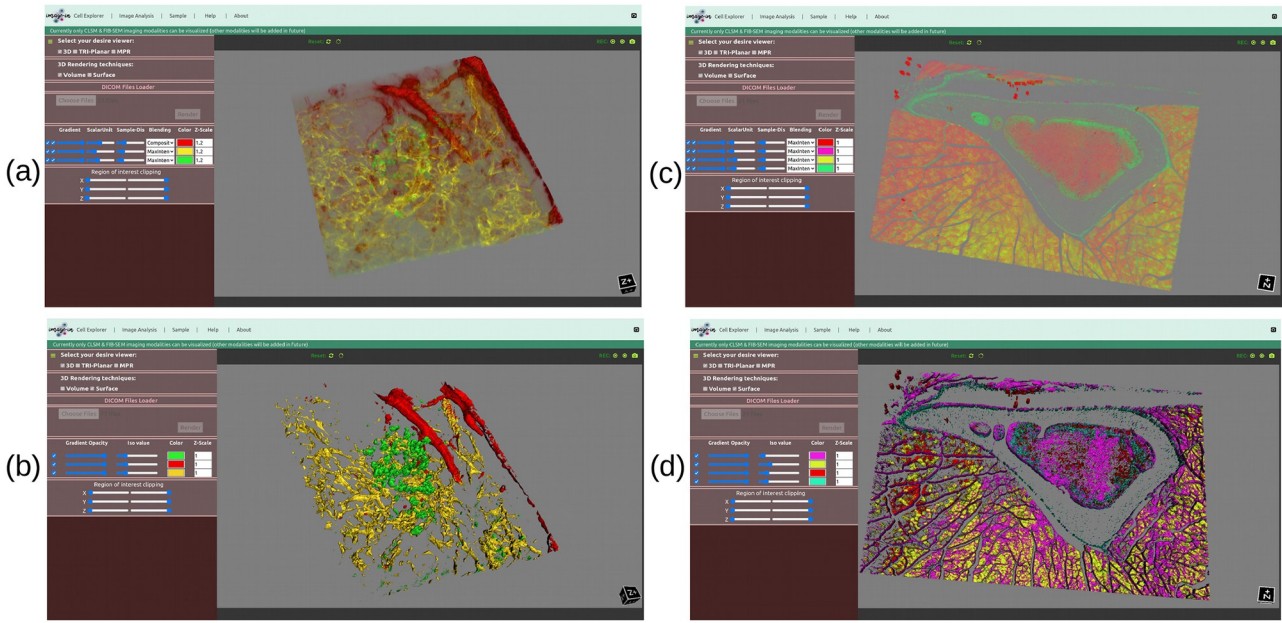

**Fig 10.** The three-channel human islet microvasculature [42] and four-channel tibia bone tissue [BL6-SEC5] [27] CLSM dataset is rendered in both (a, c) volume and (b, d) surface rendering views. In each view, each channel is rendered in a different color.

Surface rendering, on the other hand, is computationally highly expensive as compared to volume rendering since it calculates connected components within each voxel.

Fig 7(a) is a three-channel CLSM human islet microvasculature dataset whose data size is 459.6 Mb (as shown in Table 1) and rendered in volume Fig 10(a) and surface view Fig 10(b) in both systems; the time taken to load and render this image in both systems is shown in Table 2, where we can see that there is not much difference in time while loading the image samples, but there is a significant difference in rendering time (at beginning after render button is clicked), in the case of volume rendering, the laptop took 4508 ms to render it, whereas the desktop took 6995.89 ms to render it.

Similarly, in the case of surface rendering, the laptop took 20435.81 ms and the desktop took 17328.52 ms to render the isosurface, indicating that the surface rendering algorithm renders images faster in the desktop (high-performance environment) than in the laptop, possibly due to the needs of high-end hardware by the vtkMarchineCubes filter to calculate the isosurface (which works by creating cuboid-shaped cells (voxels) using the image pixels at the cube's of all eight corners). Fig 10(a) and 10(b) show the volume and surface rendered images of a three-channel CLSM human islet microvasculature microscopy image. Fig 10(a) and 10(b) additionally show that each channel has been rendered with a different color, which was made possible by the vtkColorTransfer function class for volume and the setColor property from the vtkActor class for surface rendering.

Furthermore, we pass 2.0 Gb two-channel adipose tissue [43] and 2.22 Gb four-channel tibia bone tissue [C3H-SEC9] [27] CLSM images which have 136 and 84 slices, respectively, to evaluate the performance of our proposed visualizer, and the rendered results are shown in Fig 11(a) and 11(c) for volume, and (b, d) for surface, with loading and rendering time as shown in Table 2. In terms of volume rendering time for two-channel adipose tissue [43], the laptop device took 27939.09 ms compared to the desktop device's 31247.11 ms, and in terms of

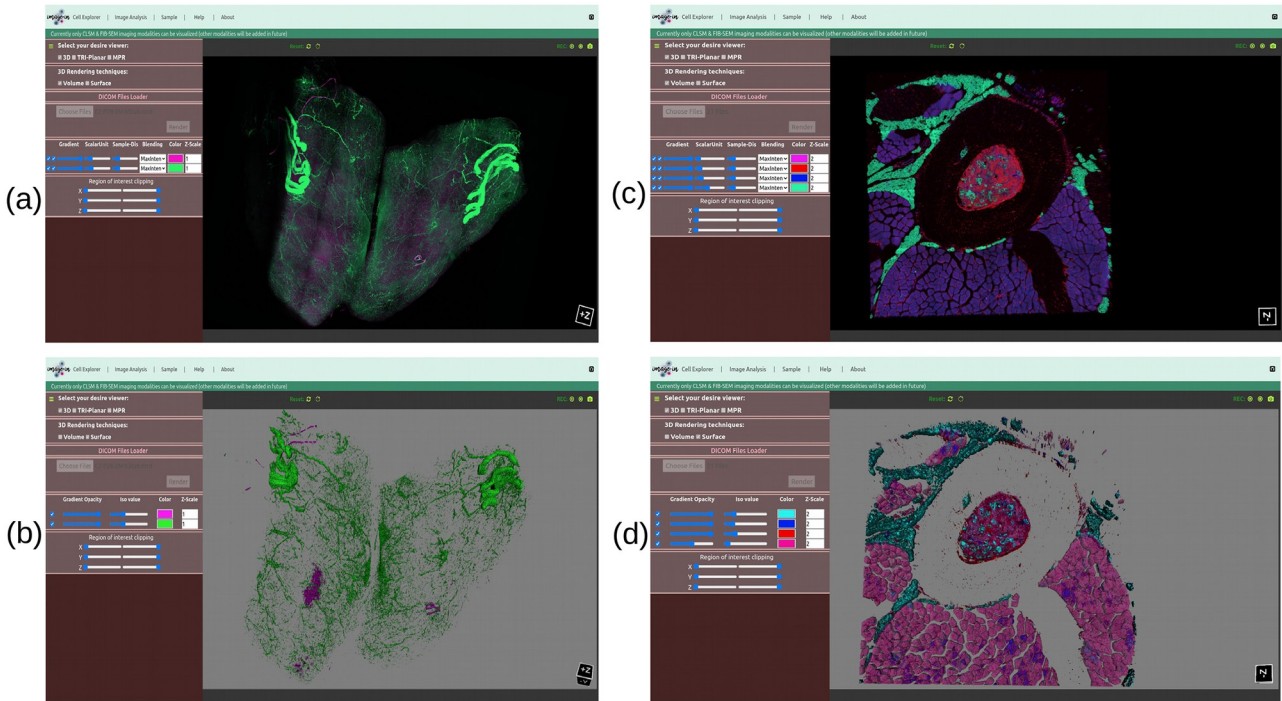

**Fig 11.** The two-channel adipose tissue [43] and four-channel tibia bone tissue [C3H-SEC9] [27] CLSM dataset is rendered in both (a, c) volume and (b, d) surface rendering views. In each view, each channel is rendered in a different color.

surface rendering, the desktop device once again outperformed the laptop device while generating an isosurface for the two-channel adipose tissue [43] CLSM image.

A similar pattern can be seen for four-channel tibia bone tissue [C3H-SEC9] [27], where in the case of volume rendering, the laptop device outperforms the desktop device, but in the case of surface rendering, the desktop device outperforms the laptop device once more while generating an isosurface for four-channel tibia bone tissue [C3H-SEC9] [27] CLSM image. Likewise, Fig 10(c) and 10(d) shows the rendered volume and surface results for four-channel tibia bone tissue [BL6-SEC5] [27] CLSM dataset, and Table 2 shows the time taken by the respective device to load and render the respective image data in a Google Chrome web browser.

Likewise, in the case of the FIB-SEM microscope modality, we pass two FIB-SEM samples, namely Plasmodium falciparum [48] which has 427 slices with a 3.9 Gb image size, and Parasitophorous vacuole [49] which has 360 slices with a 942.7 Mb image size, through the volume rendering mode to render them in 3D volumes; the obtained results are shown in Fig 12(a) and 12(b) and it's rendering, and image loading time is shown in Table 2. Furthermore, we run the Tuwongella immobilis [47] FIB-SEM image through the tri-planar viewer to render it into three planes (XY, XZ, YZ), as shown in Fig 12(c) and 12(d) shows the multiplanar reconstruction view of the Plasmodium falciparum [48] FIB-SEM image, where each plane is orthogonal to each other.

In addition, we tested the IMAGE-IN web-viewer performance on the iPad Air (3rd generation, 64 Gb). Because the computational graphical power of the iPad is limited, we pass three small data sets (three-channel mouse proximal colon-8 (147.7 Mb) [44], two-channel submandibular ganglion-9 (68.2 Mb) [45], and two-channel mouse proximal colon-10 (297.8 Mb) [46]) to measure the loading and rendering time performance on the iPad device, and later we

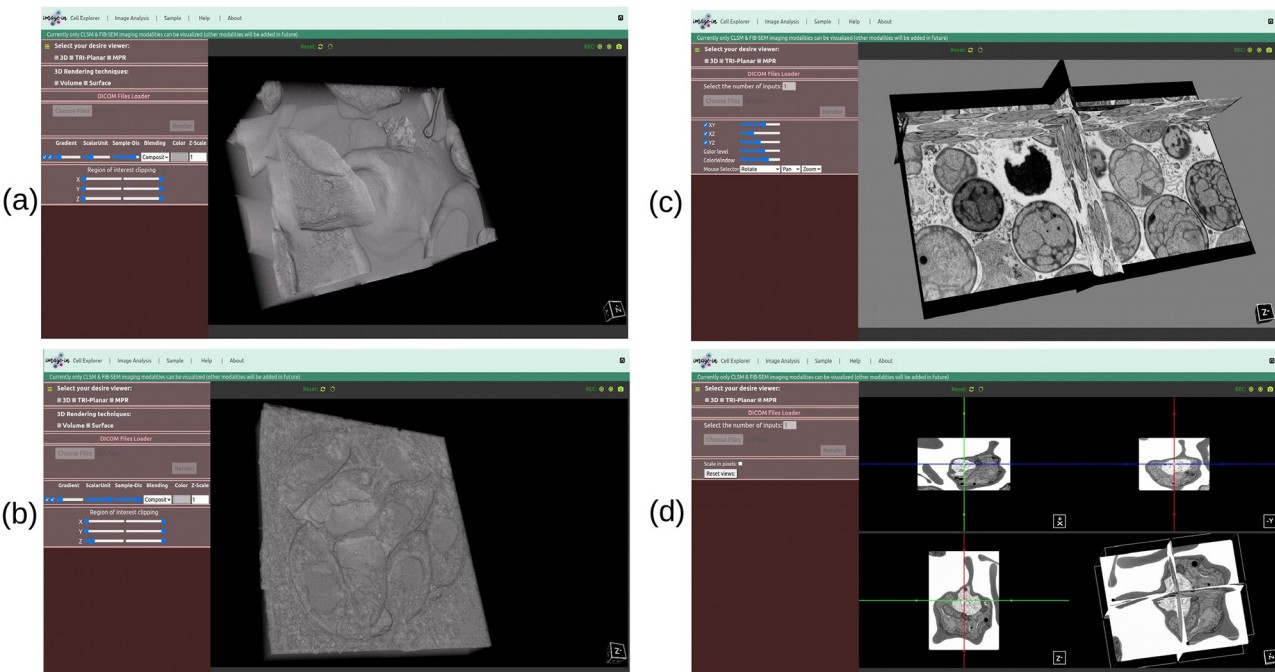

**Fig 12.** (a) and (b) show the volume rendering of Plasmodium falciparum [48] and Parasitophorous vacuole [49] FIB-SEM microscope imaging, respectively, whereas (c) shows the tri-planar view of Tuwongella immobilis [47], and (d) shows the multiplanar reconstruction view of Plasmodium falciparum [48] FIB-SEM microscope imaging modality.

pass the same three data sets from laptop and desktop for comparison purposes, as shown in Table 2. Table 3 shows the time taken by the iPad device to load and render the respective (three-channel mouse proximal colon-8 (147.7 Mb) [44], two-channel submandibular ganglion-9 (68.2 Mb) [45], and two-channel mouse proximal colon-10 (297.8 Mb) [46]) image data in a Google Chrome web browser.

For the three-channel mouse proximal colon-8 [44] data, the iPad device took 2273 and 7069 ms to render it into volume and surface, as shown in Table 3, which are high compared to laptop and desktop performance for the same input (as shown in Table 2). Moreover, in the case of two-channel submandibular ganglion-9 (68.2 Mb) [45], a similar pattern can be noticed; the iPad device requires more computing power to render them into volume and surface than the other two devices. Furthermore, in the case of volume rendering for the two-channel mouse proximal colon-10 (297.8 Mb) [46], iPad device took 2788 ms, which is again excessive when compared to laptop (2086.6 ms) and desktop (2275.75 ms) devices. When the

**Table 3. 3D viewer performance measurement on Ipad device.**

| Samples | Ipad device | | | |
|---|---|---|---|---|
| | Loading time | | Render time | |
| | VOL (ms) | SUR (ms) | VOL (ms) | SUR (ms) |
| Mouse-8 [44] | 27.32 | 29.31 | 2273 | 7069 |
| Submandibular-9 [45] | 25.54 | 26.30 | 1055 | 1727 |
| Mouse-10 [46] | 32.41 | . | 2788 | . |

VOL: Volume rendering, SUR: Surface rendering, ms: Millisecond.

same two-channel mouse proximal colon-10 (297.8 Mb) [46] data was surface rendered, the iPad device web-browser began to crash. This is because the surface rendering algorithm demands a lot of computing power to compute marching cubes, and this also happens with larger data sets. However, the same two-channel mouse proximal colon-10 (297.8 Mb) [46] successfully renders onto the surface in the case of laptop and desktop devices, and their loading and rendering are shown in Table 2.

In the case of the iPad device, the rendered image quality and interaction smoothness were satisfactory in all viewers for all small datasets, with the exception of a slight lag in the case of volume rendering in composite blend mode for the two-channel mouse proximal colon-10 (297.8 Mb); nevertheless, for large datasets, such as those > 300 Mb, we observed a web-browser crash in the case of surface and volume rendering.

## Feature discussion

In our viewer, we have included the features as suggested by our project collaborators with ample experience in bioimaging and image visualization, especially utilizing confocal and FIB-SEM microscope datasets.

According to our discussions with collaborators from the field of microbiology, we realize that a publicly available GPU-based 3D rendering viewer is needed because the increasingly large datasets are hard to visualize with their fine details. The user normally wants to explore the dataset quickly and frequently wants to maintain high rendering quality throughout viewport interaction, so that he or she can keep track of key structural elements. Therefore, in this task-2, we have built an interactive web-based 3D visualizer that leverages the WebGL graphics (client CPU/GPU device) rendering algorithm to render a given input image into volume, surface, tri-planar, and MPR viewing modes. This feature enables the user to fine-tune the rendering parameters in real-time and receive visible feedback instantly without having to wait for the changes to take effect. Of course, this capability is dependent on the user device's hardware capacity and graphics maximum buffer size.

As previously stated, our web viewer was optimized to display confocal and FIB-SEM datasets. CLSM data are often multichannel, whereas FIB-SEM are typically single-colour channel data (or a series of slices). Most available viewers do not pay much attention to how to display or interpret multichannel CLSM images interactively in one viewport while clearly displaying individual channels or relationships between them. However, our proposed interactive web-based 3D visualizer can display multichannel CLSM datasets (each channel with a different colour) in a single viewport efficiently. Figs 10 and 11 show the outcomes of a multichannel CLSM image displayed in our proposed 3D visualizer. Fig 12(a) and 12(b) presents the outcomes of a FIB-SEM image rendered in our proposed 3D volume viewer.

Furthermore, in the case of our 3D volume rendering modes, we have implemented various blending techniques to make the generated volume image understandable. This capability comes in handy when dealing with multichannel data. The user can switch between modes inside the same viewport to better grasp the spatial connections between each channel. Likewise, both the volume and surface 3D viewers have an interactively built opacity slider that allows the user to reduce or increase the rendered image transparency. This feature was made interactive by first calculating the sampled volume data points along the projected rays, then gathering their visual structure using a transfer function that converts data values to color and opacity values, and finally putting these obtained functions in an interactive window slider where a user can change their values to emphasize different structures or aspects of the provided data. This capability comes in handy as well when the viewport has rendered more than one image sample. Similarly, we have added the region of interest clipping approaches to assist

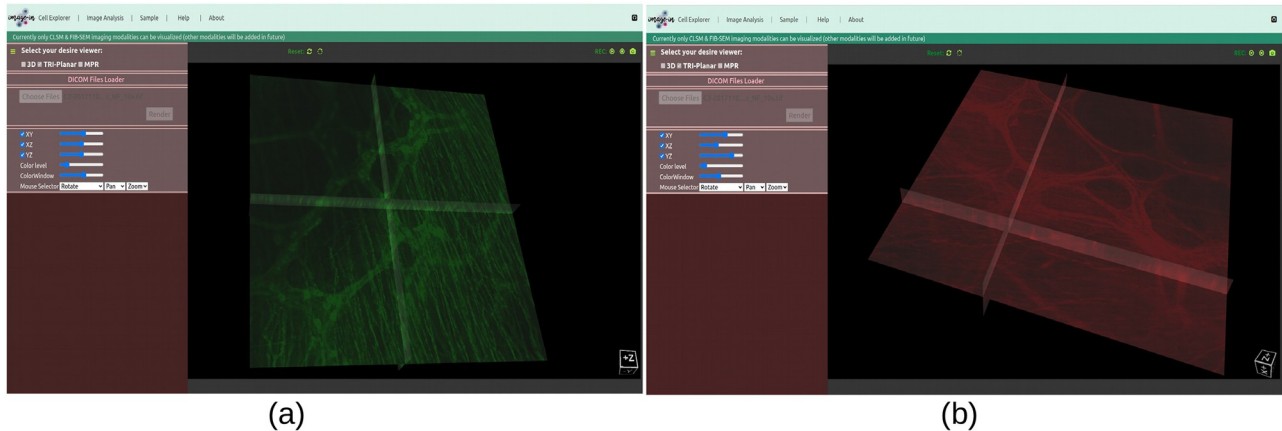

<div style="text-align: center;">(a)                 (b)</div>

**Fig 13. The two-channel mouse proximal colon [46] CLSM dataset is rendered into tri-planar viewer.** (a) Channel-1: (green) neuronal nitric oxide synthase (nNOS) (b) Channel-2: (red) neurofilament (NF) heavy chain.

users in trimming the rendered undesired region when analyzing the 3D rendered image. This option is available in both the volume and surface rendering viewports.

Edge visibility features have also been added to the volume and surface modes of display to provide users additional control over the rendered image's edge. This feature allows the user to switch on/off the edge lighting of a given rendered image. To change the scale of the generated image, a z-voxel dimension scaling factor has been implemented in both the volume and surface display modes, and this functionality will come in handy when the rendered image does not have enough slices.

Similarly, in the case of the tri-planar viewer, we have included three cross-sectional views, namely, XY, XZ, and YZ axes, and each axis has visibility on/off features, which allows microbiologists to perform image analysis based on their preferred axis, as shown in Fig 13 where we have rendered a two-channel mouse proximal colon [46] in a tri-planar viewer. This viewer also has interactive mouse selector features such as zooming, rotation, and panning. Likewise, in the case of the MPR web viewer, we have implemented three orthogonal plane viewers, namely, axial, coronal, and sagittal, to aid microbiologists in exploring insights while undertaking deep analysis of the volume data. Both tri-planar and MPR viewers provide window and colour level increase and decrease features that help to maintain the contrast and brightness of the rendered data.

In addition, we have implemented capabilities such as a video recorder, snapshot, canvas reset, and canvas background colour change to assist researchers in making better use of rendered images.

Overall, as demonstrated in Table 2, our proposed interactive multidimensional viewer performed very well when rendering both small and large data into a volume or in a surface viewer.

## Conclusion

In this article, we present IMAGE-IN, an interactive web-based multidimensional 3D visualizer that accepts a sequence of files or stacked 3D image files from multichannel CLSM and FIB-SEM microscope imaging as input and renders them into volume, surface, tri-planar, and MPR views based on the user's preferences. This viewer was developed in response to our

microbiologist collaborators' need to display multicellular multidimensional data in their preferred web browser. Each proposed viewer has its own set of features and benefits.

The scientific visualization software VTK.js was used to render 3D objects in a web browser. Our proposed 3D visualizer uses WebGL graphic acceleration features (if the system supports them) to render images in high resolution and to make the interactive response time quick. Available 3D visualization tools were deemed insufficient to visualize all channels of a multi-channel CLSM image together, however, in our visualizer, it's possible to render multichannel CLSM images together, and each rendered channel has been represented with a different colour as shown in Figs 10 and 11. Likewise, Fig 12 shows the FIB-SEM image rendered into volume, tri-planar, and MPR modes.

IMAGE-IN's performance is evaluated by observing the load and first render times on two different systems (laptop and desktop). Table 1 presents the selected input data, whereas Table 2 shows the performance evaluation. Similarly, we compared the rendered image quality of our proposed IMAGE-IN viewer to that of a publicly available ClearVolume, Napari, and a privately available (closed-source) Imaris viewer, and the results were almost identical, as shown in Figs 8 and 9.

In the future, we would like to work on temporal data sequences as well as the integration of a physically based rendering (PBR) algorithm in our viewer. It would also be interesting to incorporate a deep learning segmentation pipeline into our viewer so that users can automatically separate their data based on learning models.

## Acknowledgments

We thank the VTK.js developers and community members (https://discourse.vtk.org/) for their helpful suggestions on the issues we encountered when developing an IMAGE-IN viewer.

## Author Contributions

**Conceptualization:** Yubraj Gupta.

**Formal analysis:** Yubraj Gupta, Carlos Costa, Eduardo Pinho, Rainer Heintzmann.

**Investigation:** Yubraj Gupta.

**Methodology:** Yubraj Gupta.

**Project administration:** Carlos Costa, Eduardo Pinho, Rainer Heintzmann.

**Resources:** Yubraj Gupta.

**Software:** Yubraj Gupta.

**Supervision:** Carlos Costa, Eduardo Pinho, Luís A. Bastião Silva, Rainer Heintzmann.

**Validation:** Yubraj Gupta, Eduardo Pinho, Luís A. Bastião Silva, Rainer Heintzmann.

**Visualization:** Yubraj Gupta, Carlos Costa, Eduardo Pinho, Luís A. Bastião Silva, Rainer Heintzmann.

**Writing – original draft:** Yubraj Gupta.

**Writing – review & editing:** Yubraj Gupta, Carlos Costa, Eduardo Pinho, Rainer Heintzmann.

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
