## [Decision Letter · Decision Letter 0]

14 Nov 2022

PONE-D-22-25990IMAGE-IN: Interactive Web-based Multidimensional 3D Visualizer for Multi-Modal Microscopy ImagesPLOS ONE

Dear Dr. Gupta,

Thank you for submitting your manuscript to PLOS ONE. After careful consideration, we feel that it has merit but does not fully meet PLOS ONE’s publication criteria as it currently stands. Therefore, we invite you to submit a revised version of the manuscript that addresses the points raised during the review process.

We look forward to receiving your revised manuscript.

Kind regards,

Talib Al-Ameri, Ph.D

Academic Editor

PLOS ONE

Journal Requirements:

3. PLOS requires an ORCID iD for the corresponding author in Editorial Manager on papers submitted after December 6th, 2016. Please ensure that you have an ORCID iD and that it is validated in Editorial Manager. To do this, go to ‘Update my Information’ (in the upper left-hand corner of the main menu), and click on the Fetch/Validate link next to the ORCID field. This will take you to the ORCID site and allow you to create a new iD or authenticate a pre-existing iD in Editorial Manager. Please see the following video for instructions on linking an ORCID iD to your Editorial Manager account: https://www.youtube.com/watch?v=_xcclfuvtxQ.

4. Please review your reference list to ensure that it is complete and correct. If you have cited papers that have been retracted, please include the rationale for doing so in the manuscript text or remove these references and replace them with relevant current references. Any changes to the reference list should be mentioned in the rebuttal letter that accompanies your revised manuscript. If you need to cite a retracted article, indicate the article’s retracted status in the References list and also include a citation and full reference for the retraction notice.

 Reviewers' comments:

Reviewer's Responses to Questions

**Comments to the Author**

1. Is the manuscript technically sound, and do the data support the conclusions?

Reviewer #1: Yes

Reviewer #2: Yes

2. Has the statistical analysis been performed appropriately and rigorously? 

Reviewer #1: N/A

Reviewer #2: Yes

3. Have the authors made all data underlying the findings in their manuscript fully available?

Reviewer #1: Yes

Reviewer #2: Yes

4. Is the manuscript presented in an intelligible fashion and written in standard English?

Reviewer #1: Yes

Reviewer #2: Yes

5. Review Comments to the Author

Reviewer #1: � The purposes and set-up of this project have been built properly

In the methodology for IMAGE_IN comparison, I hoped to use a commonly used close source 3D rendering application beside ImageJ, the open one.

I did not get why in the results they have focused on IMAGE_IN performance on laptop and desk top comparison on loading and render time. Maybe if the comparison was about IMAGE_IN performance between laptop and mobile systems.

I found the tri-planner and the multi-planner viewer that are provided with this system is of great privilege of IMAGE-IN when compared for example to ImageJ application, yet I did not see that the authors have highlighted the advantages of having them.

Comparison between IMAGE-IN, ClearVolume, and Napari viewer, is the best part in results and discussion section maybe moving it to the front of the topic is better

What I found not handy while using IMAGE-IN is the necessity of converting the images into DICOM format.

I believe that IMAGE-IN web will be of great value if it is more flexible regarding data format like lif rather than accepting DICOM only.

Reviewer #2: Thank you for submitting your manuscript.

The authors introduced a web-based 3D visualizer for microscope images. They compared their method using a laptop and desktop, and they compared the images produced by the visualizer with other software programs.

The topic is really interesting, the authors developed a user-friendly visualizer that helps researchers to access the software from PCs and eliminates the use of commercial software.

However, I have the following comments that can improve the quality of this work

- The spelling of the short title must be corrected ‘’ IMAGE-IN multidimensisional web viewer’’

- Lines 98,99,100 are repetition.

- Table 2 presents the performance of the software using laptop and desktop, the results were presented in msec, how did you eliminate the effect of internet services?

It would be really helpful if the comparison was after a time interval rather than laptop and desktop so you can measure the reliability of the software.

6. PLOS authors have the option to publish the peer review history of their article (what does this mean?). If published, this will include your full peer review and any attached files.

Reviewer #1: No

Reviewer #2: No

---

## [Author Response · Author response to Decision Letter 0]

5 Dec 2022

Editor/Journal Requirements:

-- Thank you for your comments. We wrote an article using the PLOS ONE latex template. With

respect to code sharing, we have provided GitHub links as well as a DOI link in the reference

section: (https://doi.org/10.5281/zenodo.7351555). In the editorial manager, the corresponding

author registered his ORCID ID. We double-checked the reference section and found no errors. We

added one extra reference to our manuscript since we obtained one image from the Sparc database

for our study and so needed to cite the data provider. And lastly, we have processed all figures using

the Preflight Analysis and Conversion Engine (PACE) digital diagnostic tool to adhere to the PLOS

One standard image in .tif files.

“Yuan PQ, Wang L, Mulugeta M, Tache Y. CLARITY and three-dimensional (3D) imaging of the mouse and

porcine colonic innervation; 2022. Available from: https://sparc.science/datasets/31/version/4.”

Reviewer 1:

- In the methodology for IMAGE_IN comparison, I hoped to use a commonly used close

source 3D rendering application beside ImageJ, the open one.

-- Thanks for the suggestion to compare both to open-source and commercial software. We

therefore now compare the presented IMAGE-IN web viewer to ClearVolume (the ImageJ open-

source desktop viewer), Napari (the Python desktop viewer), and Imaris (the closed-source desktop

viewer).

We changed the text (line numbers 425–449) and Figures 8 and 9 show the results:

“Comparison between IMAGE-IN, ClearVolume, Napari, and Imaris viewers”

“In this section, we compared the results acquired from IMAGE-IN to the publicly available ClearVolume (a

renderer running inside ImageJ) and Napari (a renderer running inside Python) viewers, as well as with the

commercial Imaris (a closed-source desktop viewer) viewer. Figs 8 and 9 show the comparative results of

each visualizer when two-channel submandibular ganglion [45] and three-channel mouse proximal colon

[44] CLSM microscope images were passed through them. Aside from modifying mouse manipulators such

as zoom and panning, these image screenshots were captured without adjusting any parameters. IMAGE-IN,

ClearVolume, and Imaris viewers provide functionality such as region of interest clipping, voxel dimension

scaling, an opacity slider, a color picker, and more. However, the native Napari Python visualizer does not

provide region of interest clipping or voxel dimension scalings. ClearVolume only supports maximum

intensity projection to render volumetric data; however, our visualizer offers three different blending modes

to render volumetric data: composite (default), maximum, and minimum intensity projection. Napari offers

five blending modes: additive, average, maximum (selected by default), and minimum intensity projection.

Imaris, for example, offers four blending modes: MIP (maximum intensity projection), normal shading,

Blend, and shadow projection. To render the provided input CLSM data, we use the maximum intensity

projection mode in all three visualizers (Figs 8 and 9).Our visualizer provides surface (3D isosurface), tri-planar (cross-sections), and MPR (orthogonal sections)

viewers. The rendered image quality is nearly identical in all viewers. The user interface (UI) in IMAGE-IN

is substantially quicker than in Napari and ClearVolume viewer. However, the commercial (closed source)

Imaris viewer is faster than all three viewers.”

- I did not get why in the results they have focused on IMAGE_IN performance on laptop and

desktop comparison on loading and render time. Maybe if the comparison was about

IMAGE_IN performance between laptop and mobile systems.

-- The IMAGE-IN web viewer's rendering speed and image quality heavily rely on the client's

(frontend side) computer power. This is due to the IMAGE-IN design, in which the backend is

solely needed to run the Django web framework (server) in order to host the webpage; the backend

has no impact on loading and rendering the data, which is entirely handled on the frontend. To give

a realistic idea about the rendering limit on a typical desktop and laptop, we, therefore, measured

and presented their rendering performance.

Thanks for suggesting to also compare to mobile devices. We now also present IMAGE-IN

rendering times on an iPad Air (4th generation) in Table 3.

Changed text here: line numbers 535–564.

“In addition, we tested the IMAGE-IN web-viewer performance on the iPad Air (3rd generation, 64 Gb).

Because the computational graphical power of the iPad is limited, we pass three small data sets (three-

channel mouse proximal colon-8 (147.7 Mb) [44], two-channel submandibular ganglion-9 (68.2 Mb) [45],

and two-channel mouse proximal colon-10 (297.8 Mb) [46]) to measure the loading and rendering time

performance on the iPad device, and later we pass the same three data sets from laptop and desktop for

comparison purposes, as shown in Table 2. Table 3 shows the time taken by the iPad device to load and

render the respective (three-channel mouse proximal colon-8 (147.7Mb) [44], two-channel submandibular

ganglion-9 (68.2 Mb) [45], and two-channel mouse proximal colon-10 (297.8 Mb) [46]) image data in a

Google Chrome web browser. For the three-channel mouse proximal colon-8 [44] data, the iPad device took

2273 and 7069 ms to render it into volume and surface, as shown in Table 3, which are high compared to

laptop and desktop performance for the same input (as shown in Table 2). Moreover, in the case of two-

channel submandibular ganglion-9 (68.2 Mb) [45], a similar pattern can be noticed; the iPad device requires

more computing power to render them into volume and surface than the other two devices. Furthermore, in

the case of volume rendering for the two-channel mouse proximal colon-10 (297.8 Mb) [46], iPad device

took 2788 ms, which is again excessive when compared to laptop (2086.6 ms) and desktop (2275.75 ms)

devices. When the same two-channel mouse proximal colon-10 (297.8 Mb) [46] data was surface rendered,

the iPad device web-browser began to crash. This is because the surface rendering algorithm demands a lot

of computing power to compute marching cubes, and this also happens with larger data sets. However, the

same two-channel mouse proximal colon-10 (297.8 Mb) [46] successfully renders onto the surface in the

case of laptop and desktop devices, and their loading and rendering are shown in Table 2.

In the case of the iPad device, the rendered image quality and interaction smoothness were satisfactory in all

viewers for all small datasets, with the exception of a slight lag in the case of volume rendering in composite

blend mode for the two-channel mouse proximal colon-10 (297.8 Mb); nevertheless, for large datasets, such

as those > 300 Mb, we observed a web-browser crash in the case of surface and volume rendering.”

- I found the tri-planner and the multi-planner viewer that are provided with this system is of

great privilege of IMAGE-IN when compared for example to ImageJ application, yet I did not

see that the authors have highlighted the advantages of having them.

-- Thank you for stressing this unique selling point. We liked the idea and therefore now highlighted

the features of the tri-planar and MPR viewers in line numbers 612–621.

Changed text here: line numbers 612-621

“Similarly, in the case of the tri-planar viewer, we have included three cross-sectional views, namely, XY,

XZ, and YZ axes, and each axis has visibility on/off features, which allows microbiologists to perform image

analysis based on their preferred axis, as shown in Fig 13 where we have rendered a two-channel mouse

proximal colon [46] in a tri-planar viewer. This viewer also has interactive mouse selector features such as

zooming, rotation, and panning. Likewise, in the case of the MPR web viewer, we have implemented three

orthogonal plane viewers, namely, axial, coronal, and sagittal, to aid microbiologists in exploring insights

while undertaking deep analysis of the volume data. Both tri-planar and MPR viewers provide window and

colour level increase and decrease features that help to maintain the contrast and brightness of the rendered

data.”

- Comparison between IMAGE-IN, ClearVolume, and Napari viewer, is the best part in

results and discussion section maybe moving it to the front of the topic is better.

-- We agree to this good suggestion and have moved the comparison section from line numbers 544-

567 further to the front (425-449).

- What I found not handy while using IMAGE-IN is the necessity of converting the images

into DICOM format. I believe that IMAGE-IN web will be of great value if it is more flexible

regarding data format like lif rather than accepting DICOM only.

-- This was a valid drawback of our viewer, which is why we now invested the extra effort

(Bioformats is not yet available in JavaScript) to support more file formats. The new edition of the

IMAGE-IN web viewer now supports .dcm, .tif, .tiff, .nrrd, .mha, .nii, .ome.tif, .png, .bmp, .jpg, and

some other formats; however, due to the unavailability of a generic import tool in JavaScript, the

IMAGE-IN viewer is still unable to import proprietary file formats such as .czi, .lif, .nd2, and

others. Line numbers: 263-267.

“Our IMAGE-IN web-viewer can currently render input images belonging to .dcm, .tif, .tiff, .nrrd, .mha, .nii,

.ome.tif, .png, .bmp, .jpg, and other formats; however, due to the unavailability of a generic import tool in

JavaScript, the IMAGE-IN web-viewer is still unable to load proprietary files such as .czi, .lif, .nd2.”

Reviewer 2:

- The spelling of the short title must be corrected “IMAGE-IN multidimensisional web

viewer’’.

-- Thanks for picking up this mistake, which we now corrected to "IMAGE-IN multidimensional

web viewer".

- Lines 98,99,100 are repetition.

-- Thanks for picking this up. We now have deleted the below paragraph from the manuscript (99-

103).

“Furthermore, the purpose of this study (task-2) is to provide an interactive web-based multidimensional 3D

visualizer for users that allows them to visualize any dicomized multidimensional image in a web browser

rather than on a local computer where an installer file of the targeted OS visualizer of a specific vendor must

be installed.”

- Table 2 presents the performance of the software using laptop and desktop, the results were

presented in msec, how did you eliminate the effect of internet services? It would be really

helpful if the comparison was after a time interval rather than laptop and desktop so you can

measure the reliability of the software.

-- Thanks for picking up on a potential misunderstanding about including the less reliable internet

services in our time measurements. IMAGE-IN web viewer performance has been measured after

the webpage is fully loaded on the client side, which means the internet service has no impact on

the measurement. And furthermore, we are now measuring loading and image rendering times on

three devices: a laptop, a desktop, and an iPad Air, to get an indication of how computing power

affects the end-user experience. Consequently, we are now clarifying this by reporting both

“Loading Time” and “Render Time” and changing the text to (453–468):

“Later, we evaluated the performance of the IMAGE-IN web viewer on the frontend rather than the

backend because the backend is only used to host the Django web server and all other libraries

(such as ITK.js and VTK.js) are loaded on the frontend when the user loads the webpage. As a

result, we assess the rendering performance of the IMAGE-IN web viewer on the client side to

determine the impact of the client’s computer graphical power on performance. Because of this, we

chose three different devices, the first of which is a laptop with a system specification (Memory: 16

Gb, processor: Intel Core i7-10750H CPU 2.60Hz, Graphics: NVIDIA GeForce RTX 2060, OS

type: 64-bit, max 3D Texture size: 16384), the second of which is a desktop with a system

specification (Memory: 64 Gb, processor: Intel Core i7-6700K CPU 4Hz, Graphics: NVIDIA

GeForce GTX TITAN, OS type: 64-bit, max 3D Texture size: 2048), both of which were running

the Ubuntu 20.04.4 LTS (Focal Fossa) operating system and both of which had a Google Chrome

and Firefox browser pre-installed, and the third device is an iPad Air (3rd generation, 64 Gb), to

gain a better sense of how graphical computational power will affect the end-user experience.”

-- Likewise, we also measured the viewer performance on iPad Air, which can be seen in Table 3.

Line numbers 535-564.

“In addition, we tested the IMAGE-IN web-viewer performance on the iPad Air (3rd generation, 64 Gb).

Because the computational graphical power of the iPad is limited, we pass three small data sets (three-

channel mouse proximal colon-8 (147.7 Mb) [44], two-channel submandibular ganglion-9 (68.2 Mb) [45],

and two-channel mouse proximal colon-10 (297.8 Mb) [46]) to measure the loading and rendering time

performance on the iPad device, and later we pass the same three data sets from laptop and desktop for

comparison purposes, as shown in Table 2. Table 3 shows the time taken by the iPad device to load and

render the respective (three-channel mouse proximal colon-8 (147.7Mb) [44], two-channel submandibular

ganglion-9 (68.2 Mb) [45], and two-channel mouse proximal colon-10 (297.8 Mb) [46]) image data in a

Google Chrome web browser. For the three-channel mouse proximal colon-8 [44] data, the iPad device took

2273 and 7069 ms to render it into volume and surface, as shown in Table 3, which are high compared to

laptop and desktop performance for the same input (as shown in Table 2). Moreover, in the case of two-

channel submandibular ganglion-9 (68.2 Mb) [45], a similar pattern can be noticed; the iPad device requires

more computing power to render them into volume and surface than the other two devices. Furthermore, in

the case of volume rendering for the two-channel mouse proximal colon-10 (297.8 Mb) [46], iPad device

took 2788 ms, which is again excessive when compared to laptop (2086.6 ms) and desktop (2275.75 ms)

devices. When the same two-channel mouse proximal colon-10 (297.8 Mb) [46] data was surface rendered,

the iPad device web-browser began to crash. This is because the surface rendering algorithm demands a lot

of computing power to compute marching cubes, and this also happens with larger data sets. However, the

same two-channel mouse proximal colon-10 (297.8 Mb) [46] successfully renders onto the surface in the

case of laptop and desktop devices, and their loading and rendering are shown in Table 2.

In the case of the iPad device, the rendered image quality and interaction smoothness were satisfactory in all

viewers for all small datasets, with the exception of a slight lag in the case of volume rendering in composite

blend mode for the two-channel mouse proximal colon-10 (297.8 Mb); nevertheless, for large datasets, such

as those > 300 Mb, we observed a web-browser crash in the case of surface and volume rendering.”

Thank you for your comments.

---

## [Decision Letter · Decision Letter 1]

16 Dec 2022

IMAGE-IN: Interactive Web-based Multidimensional 3D Visualizer for Multi-Modal Microscopy Images

PONE-D-22-25990R1

Dear Dr. Gupta,

We’re pleased to inform you that your manuscript has been judged scientifically suitable for publication and will be formally accepted for publication once it meets all outstanding technical requirements.

Kind regards,

Talib Al-Ameri, Ph.D

Academic Editor

PLOS ONE

Reviewers' comments:

Reviewer's Responses to Questions

**Comments to the Author**

1. If the authors have adequately addressed your comments raised in a previous round of review and you feel that this manuscript is now acceptable for publication, you may indicate that here to bypass the “Comments to the Author” section, enter your conflict of interest statement in the “Confidential to Editor” section, and submit your "Accept" recommendation.

Reviewer #1: (No Response)

Reviewer #2: All comments have been addressed

2. Is the manuscript technically sound, and do the data support the conclusions?

Reviewer #1: (No Response)

Reviewer #2: Yes

3. Has the statistical analysis been performed appropriately and rigorously? 

Reviewer #1: (No Response)

Reviewer #2: Yes

4. Have the authors made all data underlying the findings in their manuscript fully available?

Reviewer #1: (No Response)

Reviewer #2: (No Response)

5. Is the manuscript presented in an intelligible fashion and written in standard English?

Reviewer #1: (No Response)

Reviewer #2: Yes

6. Review Comments to the Author

Reviewer #1: (No Response)

Reviewer #2: (No Response)

7. PLOS authors have the option to publish the peer review history of their article (what does this mean?). If published, this will include your full peer review and any attached files.

Reviewer #1: **Yes: **Hadeel Adel Al-Lami

Reviewer #2: No

---

## [Editor Report · Acceptance letter]

20 Dec 2022

PONE-D-22-25990R1 

IMAGE-IN: Interactive Web-based Multidimensional 3D Visualizer for Multi-Modal Microscopy Images 

Dear Dr. Gupta:

I'm pleased to inform you that your manuscript has been deemed suitable for publication in PLOS ONE. Congratulations! Your manuscript is now with our production department. 

Kind regards, 

on behalf of

Dr. Talib Al-Ameri 

Academic Editor

PLOS ONE